# Microstructure and Properties of Cold Sprayed NiCrAl Coating on AZ91D Magnesium Alloy

**Xiangwei Zhao, Tianshun Dong \*, Binguo Fu, Guolu Li \*, Qi Liu and Yanjiao Li**

School of Materials Science and Engineering, Hebei University of Technology, Tianjin 300130, China; zhao13342084956@163.com (X.Z.); fubinguohao@163.com (B.F.); 15232550761@163.com (Q.L.); liyanjiao2577@163.com (Y.L.)
\* Correspondence: dongtianshun@hebut.edu.cn (T.D.); liguolu@hebut.edu.cn (G.L.)

**Abstract:** Herein, a NiCrAl coating was prepared on the AZ91D magnesium alloy by cold spraying technology. The microstructure, wear resistance, and corrosion resistance of the cold sprayed NiCrAl coating were studied and compared with two NiCrAl coatings prepared by plasma spraying. The results showed that the porosity of the two-plasma sprayed NiCrAl coatings was 3.21% and 2.66%, respectively, while that of the cold sprayed NiCrAl coating was only 0.68%. The hardness of the cold sprayed NiCrAl coating (650 $HV_{0.1}$) was higher than those of the two-plasma sprayed NiCrAl coatings (300 $HV_{0.1}$, 400 $HV_{0.1}$). In the abrasion resistance test, the cold sprayed NiCrAl coating showed a lower friction coefficient (0.346), less wear volume (3.026 $mm^3$), and superior wear resistance accordingly compared with the two-plasma sprayed NiCrAl coatings. Moreover, the scanning electron microscopy (SEM) morphology at the bottom of the wear trace of the cold sprayed NiCrAl coating showed a compact mechanically mixed layers (MML) structure, and its wear mechanism was mainly abrasive wear, with some fatigue wear. In the electrochemical test, the corrosion current density of the cold sprayed NiCrAl coating ($4.404 \times 10^{-2}$ $A \cdot cm^{-2}$) was much lower than those of two plasma sprayed coatings (25.96 $A \cdot cm^{-2}$, 26.98 $A \cdot cm^{-2}$), indicating that the cold sprayed NiCrAl coating had superior corrosion resistance. Therefore, preparing a cold sprayed NiCrAl coating is a feasible method to comprehensively improve the wear resistance and corrosion resistance of the AZ91D magnesium alloy.

**Keywords:** AZ91D magnesium alloy; cold spray; plasma spray; wear resistance; corrosion resistance

## 1. Introduction

Magnesium alloys have many excellent properties, including low density, high specific strength, good thermal conductivity, etc. Therefore, they find a wide range of applications in several industries, i.e., automotive, aerospace, machinery, 3C products, and other fields [1–3]. However, due to low hardness, the wear resistance of magnesium alloys is poor [4]. Moreover, they are chemically active, and their electrochemical potential is relatively low, so they can be readily oxidized in the air and forms a loose and porous oxide film, resulting in poor corrosion resistance.

Generally, there are three ways to improve the wear resistance and corrosion resistance of magnesium alloys: (1) Improving the microstructure and manufacturing process of the alloy [5–8]; (2) Preparing a protective layer on their surface [9–14]; (3) Designing or adjusting the application environment of magnesium alloys. Among them, thermal spraying on the surface of magnesium alloys is a relatively mature process [15–18], but it will inevitably cause significant adverse effects to the magnesium alloys substrate. In recent years, the development of cold spray technology has provided a possibility for the comprehensive improvement of the surface properties of magnesium alloys.

GDCS (gas dynamic cold spray) is a new type of coating preparation process, which is based on the principle of aerodynamics [19–22]. Because the temperature of the particles does not reach the melting point during the spraying process, the thermal impact of the

excessive spraying temperature on the substrate is avoided; due to the high-speed impact of the cold sprayed particles, not only the coating is densified, but also the oxide film on the surface of the substrate can be removed, so that a good interface bonding between the coating and the substrate can be formed. Therefore, cold spray technology can provide an effective method for surface protection of magnesium alloys. But so far, there are few comprehensive studies on the wear resistance and corrosion resistance of cold spray coatings on the surface of magnesium alloys substrate.

NiCrAl alloy has excellent corrosion resistance and oxidation resistance and is widely used in the preparation of thermal sprayed protective coatings. In this study, NiCrAl coatings were prepared on the surface of the AZ91D magnesium alloy by plasma spraying technology and cold spraying technology, respectively, whereupon, the wear resistance and corrosion resistance of the coatings were studied comparatively.

## 2. Materials and Methods

### 2.1. Material and Coating Preparation

In this study, two kinds of powders were used for spraying—one was the commercially available clad-type NiCrAl powder KF-110 (Beijing Xin Zhulian Co., Ltd., Beijing, China), which had a long strip shape and a particle size range of 20–80 μm (d50 = 42 μm); the other was NiCrAl alloy powder (gas atomized, Nangong Bole Metal Materials Co., Ltd., Xingtai, China), which was spherical in shape, with a particle size range of 10–60 μm (d50 = 25 μm). The morphology and particle size of the two powders are shown in Figure 1. Due to the limitation of the spraying process itself, the long strip KF-110 powder was not suitable for cold spraying; hence, it was only used for thermal spraying in this study (hereinafter referred to as APS-KF). While NiCrAl alloy powder was used for preparing plasma sprayed coating and cold sprayed coating (hereinafter referred to as APS-NCA and CS-NCA, respectively).

The AZ91D magnesium alloy was selected as the spraying substrate and was cut into $60 \times 30 \times 10$ mm$^3$ long specimens and Ø $30 \times 10$ mm$^3$ cylindrical specimens using wire cutting, which were used for plasma spraying and cold spraying, respectively. The chemical contents of the substrate and the two powders are listed in Table 1. After cleaning in ultrasonic acetone for 10 min, it was sandblasted with 24 mesh alumina grits and then preheated at 200 °C; subsequently, the spraying was performed immediately.

**Table 1.** Chemical composition of AZ91D, KF-110 powder, and NiCrAl powder (wt.%).

| Material | Mg | Ni | Cr | Al | Mn | Zn |
|---|---|---|---|---|---|---|
| AZ91D | Bal. | 0.01 | - | 9.3 | 0.32 | 0.95 |
| KF-110 powder (Beijing) | - | Bal. | 14.6 | 4.4 | - | - |
| NiCrAl powder (Nangong) | - | Bal. | 16 | 4 | - | - |

In this study, the thermal spray coating was prepared by plasma spraying (Army Armored Forces Engineering College of the PLA, Beijing, China), the gases were Ar and H$_2$, and the cold spray coating was prepared by the German Impact 5/11 cold spray system (Beijing Bin Peng Ying Hao Technology Co., Ltd., Beijing, China), N$_2$ was used as accelerating gas, as shown in Figure 2. The expected thickness of the coatings prepared by the two methods was 300 μm. The specific spraying parameters are shown in Table 2.

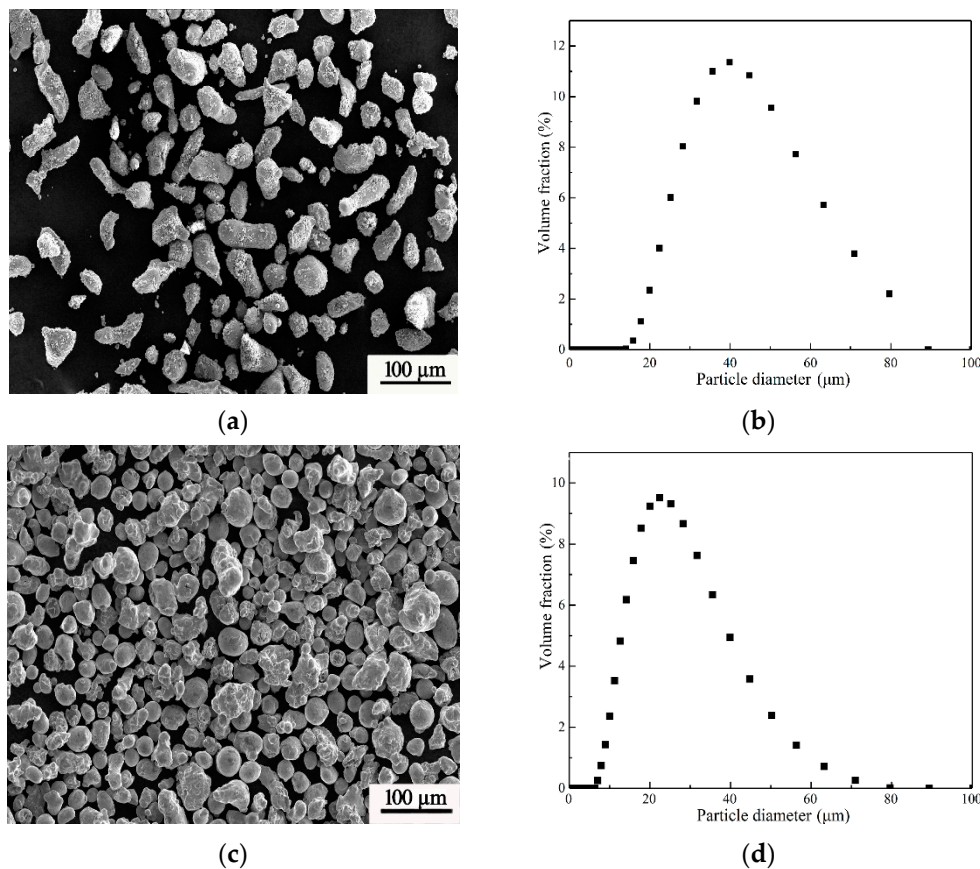

**Figure 1.** The morphology and size range of NiCrAl powder: (**a**,**b**) Coated powder; (**c**,**d**) Alloy powder.

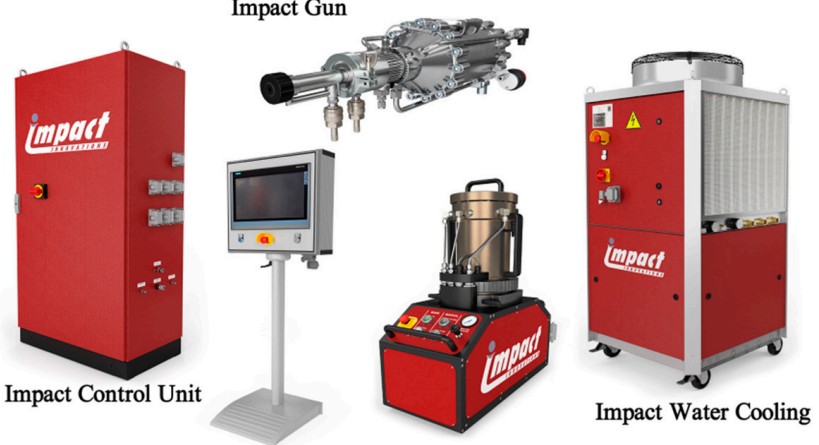

**Figure 2.** Cold spray equipment composition.

**Table 2.** Spraying parameters of the three coatings.

| Coatings Title | Power (KW) | Pressure (MPa) | Powder Feeding Rate (g/min) | Spraying Distance (mm) | Powder Feeding Amount (m$^3$/h) | Carrier Gas Flow (L/min) | Spray Gun Moving Speed (mm/s) |
|---|---|---|---|---|---|---|---|
| APS-KF | 50 | 3 | - | 110 | 10 | 16 | - |
| APS-NCA | 42 | 3 | - | 130 | 30 | 25 | - |
| CS-NCA | - | 4 | 20 | 20 | - | - | 10 |

*2.2. Characterization*

A scanning electron microscope (SEM, JSM-6510A, JEOL, Tokyo, Japan) was used to characterize the surface morphology and cross-sectional morphology of the coating. A 3D laser scanning microscope (LSM, phase shift micro XAM, KLA-tencor, Yokohama, Japan) was used to measure the average surface roughness after polishing, within the range of 0.4 mm$^2$. After testing, the surface roughness of the polished coating did not exceed Ra 0.3 μm. A Vickers microhardness tester (Hardness, DUH-211S, Shimadzu, Kyoto, Japan) was used to measure the hardness of the coating on the polished top surface (Ra < 0.3 μm) by taking the average of 5 measurements, the loading load was 980.7 mN, and the loading time was 15 s. The microhardness was also evaluated by indenting on 10 locations along the polished cross-section of the samples and doing an average of 3 measurements at each location.

*2.3. Sliding Wear Friction Test*

A UMT-3 friction and wear tester was used for the sliding wear friction test of the substrate and the coatings. The sprayed sample was cut into 10 × 10 × 10 mm$^3$ block samples. The sliding counterpart was a GCr15 steel ball, with a hardness of 62 HRC, and surface roughness did not exceed Ra 0.8 μm. The experimental exercise mode was reciprocating. The experimental parameters are listed in Table 3.

**Table 3.** Sliding wear friction test parameters.

| Lubrication Conditions | Temperature | Diameter of Steel Ball (mm) | Stroke Length (mm) | Load (N) | Test Frequency (Hz) | Experiment Time (min) |
|---|---|---|---|---|---|---|
| Dry friction | 25 °C | 6 | 4 | 10 | 10 | 20 |

*2.4. Corrosion Resistance Test*

An electrochemical workstation (CHI660, CH Instruments, Inc., Shanghai, China) was used to measure the electrochemical characteristics of the three coatings in 3.5 wt.% NaCl solution (25 °C, pH = 7). Open circuit potential (OCP), polarization curve (PDP), and electrochemical impedance spectroscopy (EIS) were tested. The sample was cut into 10 × 10 × 10 mm$^3$ and then sealed with cold mounting powder except for the coating, with an exposed area of 1 cm$^2$. Before the test, the surface of the sample was ground and polished to Ra 0.2 μm. All electrochemical tests were performed in a traditional three-electrode electrolyte battery, in which the saturated calomel electrode was used as the reference electrode (SCE), the platinum electrode as the counter electrode, and the sample as the working electrode. Under a stable open-circuit potential, a sinusoidal AC disturbance with an amplitude of 10 mV (rms) was applied in the frequency range of 0.01 Hz to 100 kHz. Finally, the polarization curve was measured at a rate of 0.5 mV/s, starting at −0.5 V/OCP up to +0.5 V/OCP. Among them, the data of PDP and EIS were analyzed with the attached program of the electrochemical workstation and Zsimp win software (ZSimpwin 3.60, Bruno Yeum, Ph.D., Ann Arbor, MI, USA).

## 3. Results and Discussion

### 3.1. Microstructure of the Coatings

Figure 3 shows the SEM morphologies of coatings. It can be seen from Figure 3a,c,e that the surface of the two plasma sprayed coatings were relatively rough, and there were some unmelted or semi-melted particles, while the surface of APS-KF coating was smoother than that of APS-NCA coating. Figure 3b,d show that there were many large cracks and pores on the cross-section of the APS-KF coating, while the cracks and pores in the APS-NCA were significantly reduced. Figure 3f shows that the porosity of the CS-NCA coating was greatly reduced, with very few cracks and pores. Image pro software was used to calculate the porosity of the three coatings according to ASTM E2109-01 [23], which were 3.21% ± 0.15%, 2.66% ± 0.11% and 0.68% ± 0.06%, respectively.

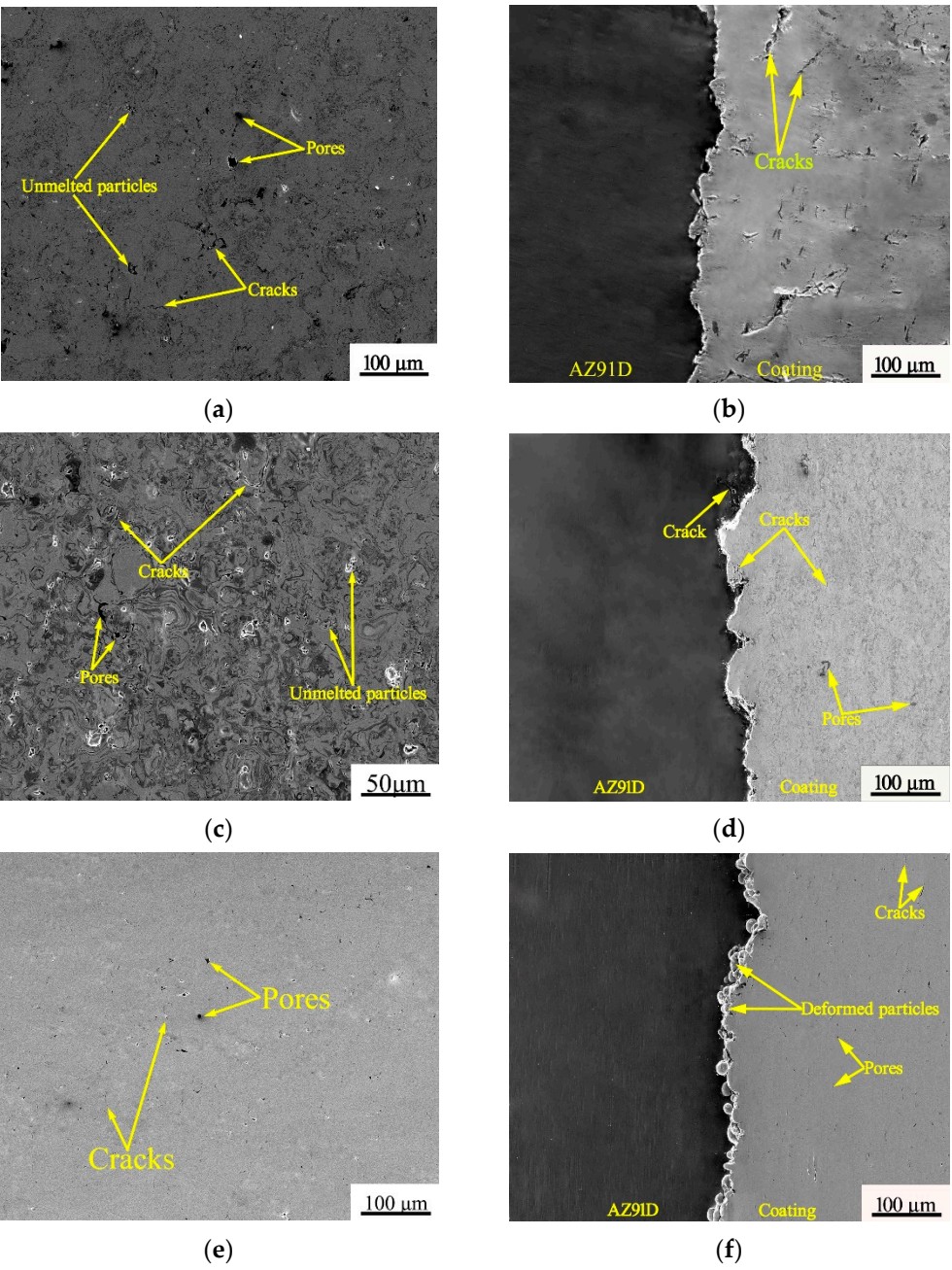

**Figure 3.** Surface and cross-sectional SEM of three coatings: (**a**,**b**) APS-KF coating; (**c**,**d**) APS-NCA coating; (**e**,**f**) CS-NCA coating.

Furthermore, it can be seen from Figure 3b,d that the interface between the two kinds of thermal sprayed coatings and the substrate showed a serrated shape, and the binding was relatively tight. Moreover, there were obvious thermal cracks on the substrate of the APS-NCA coating. Figure 3f shows that the spherical NiCrAl particles were deeply embedded in the AZ91D magnesium alloy substrate, and some particles deformed under high speed and high pressure; moreover, no defects were present at the interface.

### 3.2. Mechanical Properties

The microhardness distribution curves on the coatings cross-sections are shown in Figure 4. The average value of the microhardness of the APS-KF coating surface was about 300 $HV_{0.1}$, and the cross-section hardness distribution was relatively uniform. The average microhardness of the surface of the APS-NCA coating was about 400 $HV_{0.1}$, which was 33% higher than that of the APS-KF coating; moreover, the microhardness of the cross-section was uneven. For the CS-NCA coating, the average microhardness of the surface was 650 $HV_{0.1}$, which was more than doubled compared to the APS-KF coating. In particular, the microhardness of the CS-NCA coating tended to increase as it approached the interface. This could be attributed to the nature of cold spraying. The impact of particles resulted in large residual compressive stress at the bottom of the coating, and it was more likely to produce internal dislocations in the particles, leading to the higher hardness of the bottom of the cold spray coating.

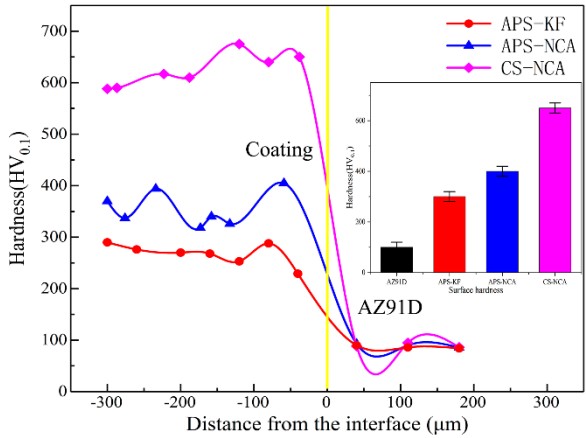

**Figure 4.** Hardness distribution curve on the cross-section of coatings. Inset is the surface microhardness.

### 3.3. Wear Performance

The friction coefficient curves of the coatings and the substrate are shown in Figure 5. As can be seen in the figure, the friction coefficient of the substrate AZ91D was the lowest, and after about 50 s of the "running in" stage, the friction coefficient showed a gradual upward trend. After 1000 s, it dropped slightly and reached a stable around 0.255.

In general, all of the friction coefficients of the three coatings showed a rapid rise to a higher level in a short time, then a rapid decline; the APS-KF coatings entered the stable wear stage after about 400 s, while the APS-NCA coating and CS-NCA coating entered the stable wear stage after about 200 s. The friction coefficients of the two plasma sprayed coatings were very close, but the friction coefficient of the APS-KF coating fluctuated very little, while that of the APS-NCA coating fluctuated greatly. This was because the surface microscopic morphology of the APS-NCA coating was rough and uneven, resulting in a large fluctuation of the friction coefficient. Compared with the two plasma spray coatings, the friction coefficient of the CS-NCA coating was obviously lower, which was about 0.346 in the stable stage, and the fluctuation was not large.

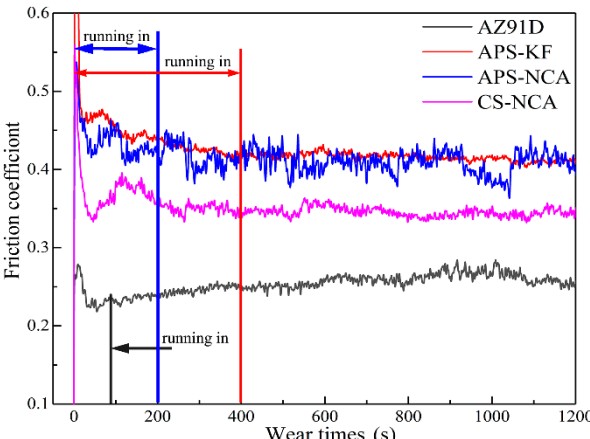

**Figure 5.** Friction coefficient curve of substrate and coatings.

Figure 6 is a schematic diagram of the three-dimensional morphology of the coatings and the substrate. There were many parallel plows at the bottom of the wear trace of AZ91D. The wear mark of the APS-KF coating had vertical edges, and the whole wear trace was very rough. The wear trace of APS-NCA coating was more vertical at the edge, and the wear trace was narrow and shallow. The wear trace of CS-NCA coating was narrower and shallower, and the width and depth of the wear trace were uneven. This could be attributed to the plastic deformation characteristic of the sprayed particles in cold spraying, which led to the uneven bonding force between the particles. During the wear process, when the radial force generated by the steel ball on the substrate was greater than the bonding force between the particles, it would cause the spray particles to delaminate. On the contrary, the particles would not fall off.

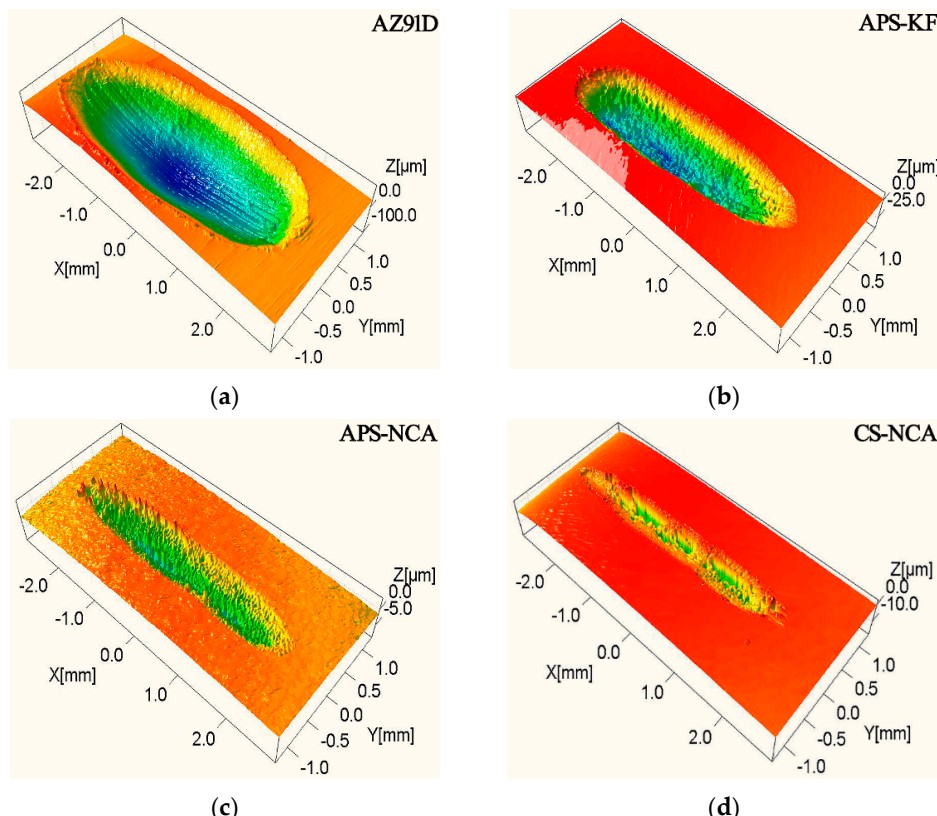

**Figure 6.** Three-dimensional topography of the worn surface: (**a**) substrate; (**b**) APS-KF coating; (**c**) APS-NCA coating; (**d**) CS-NCA coating.

Figure 7 is a statistical histogram of the wear volume and friction coefficient of the coatings and the substrate. It can be seen that the friction coefficient of the AZ91D magnesium alloy was the lowest; nevertheless, its wear volume was the largest (498.4 mm$^3$) due to low hardness. The wear volume of the APS-KF coating was 8.85 mm$^3$, while the APS-NCA coating and the CS-NCA coating were similar (4.698 and 3.026 mm$^3$, respectively). The results showed that the three kinds of coatings could effectively improve the wear resistance of the substrate, and the CS-NCA coating showed the best wear resistance.

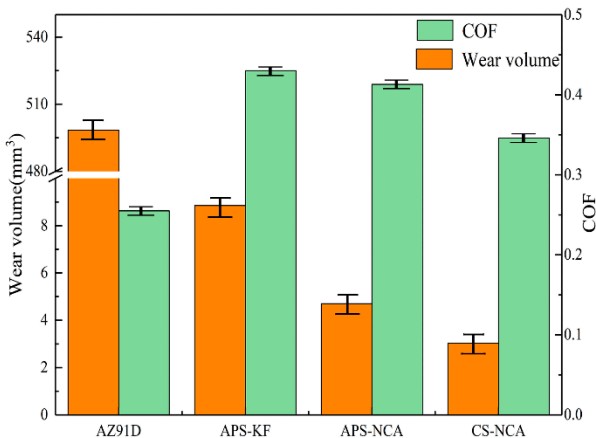

**Figure 7.** The friction coefficient and wear volume of substrate and coatings.

In order to further analyze the wear mechanism of the coatings and the AZ91D magnesium alloy, the morphology of the wear trace was observed by SEM, as shown in Figure 8. It can be seen from Figure 8a,b that there were parallel furrows and a little adhesion trace on the wear trace of the AZ91D magnesium alloy, with some cracks distributing on the edge of the furrows, and some debris were observed. Hence, the wear mechanism of AZ91D was typical abrasive wear and adhesive wear [24–27].

For the APS-KF coating, as shown in Figure 8c,d, a large amount of delamination and a few cracks could be observed, and white fractures appeared at the edge of the peeling, indicating the characteristics of fatigue wear. This was because there were many pores and cracks in the APS-KF coating (as shown in Figure 3b); during the friction process, these cracks tended to propagate parallel to the coating surface under compressive stress applied to the steel ball. Eventually, delamination occurred. Therefore, the wear mechanism of the APS-KF coating was fatigue wear.

Figure 8e,f show that the APS-NCA coating also had a similar phenomenon to the APS-KF coating, but no large-scale delamination was observed. Although it also showed some characteristics of fatigue wear, the alloy powder melted uniformly during the spraying process, leading to the particles being well-bonded; moreover, due to the higher hardness of the APS-NCA coating, the deformation was smaller under the action of the compressive stress of the steel ball. Thus, the wear of the APS-NCA coating was less than that of the APS-KF coating, and its wear mechanism was mainly fatigue wear.

Compared with the two thermal sprayed coatings above, the SEM morphology of wear trace of the CS-NCA coating showed a typical mechanically mixed layer (MML) structure [28–31], and a few of delamination could be observed, showing the characteristics of three-body abrasive wear and fatigue wear. This was perhaps because, in the sliding wear process, the coating and the steel ball would produce plastic deformation, material mixing, and abrasive debris. After a period of time, under the action of applied load, shear stress, and chemical reaction (generally oxidation reaction), the wearing debris would be pressed on the surface of the coating, forming a protective mechanical mixing layer. However, as there was no hard phase in MML structure to play the role of "support", MML removal would occur, showing the delamination phenomenon of fatigue wear. Therefore, the wear mechanism of the CS-NCA coating was three-body abrasive wear and a few

fatigue wears. Meanwhile, the CS-NCA coating had a low friction coefficient and the least amount of wear.

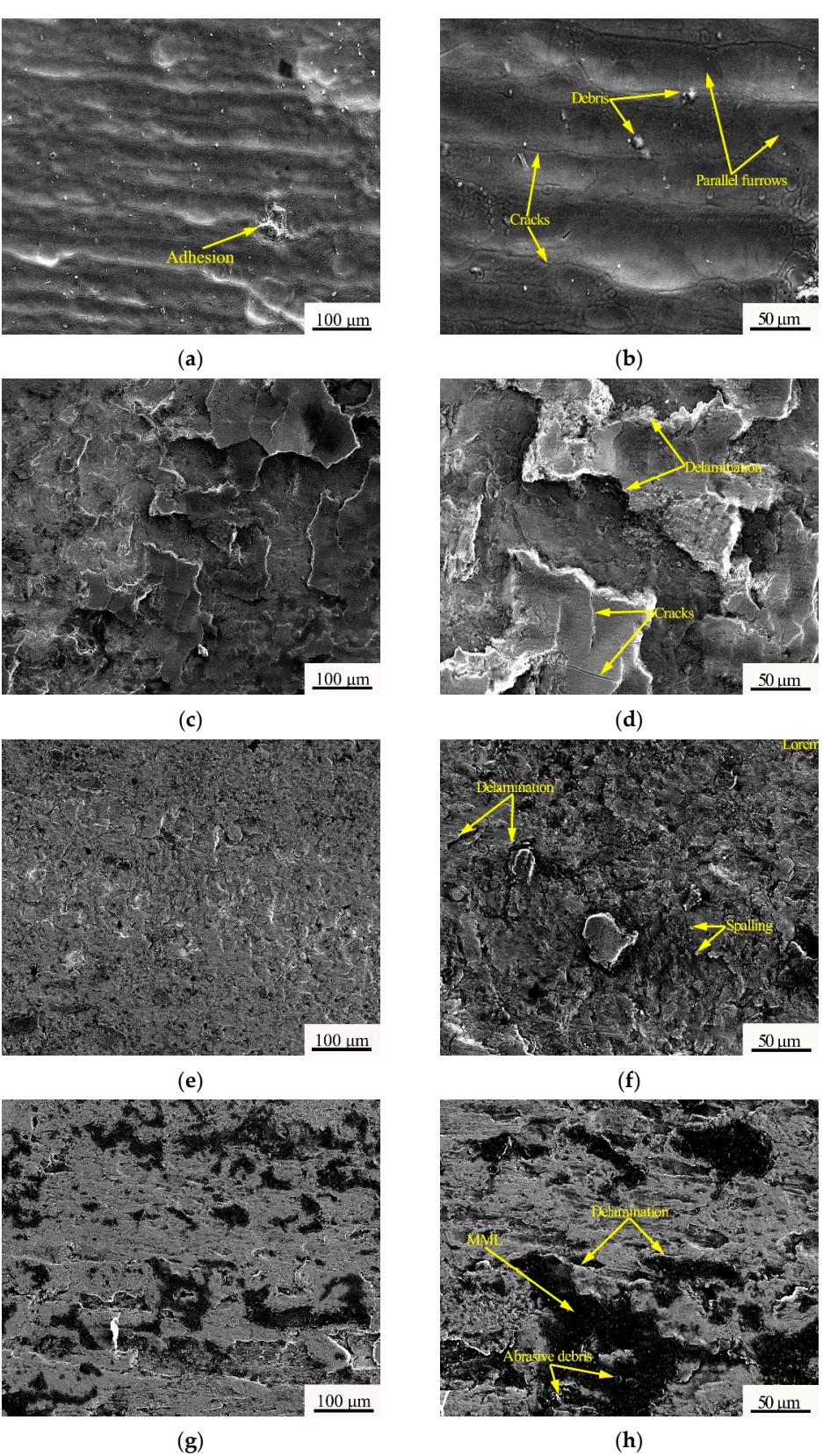

**Figure 8.** Low and high magnification scanning morphologies of the substrate and three coatings: (**a**,**b**) AZ91D magnesium alloy; (**c**,**d**) APS-KF coating; (**e**,**f**) APS-NCA coating; (**g**,**h**) CS-NCA coating.

The sliding wear friction test results above demonstrated that the three kinds of NiCrAl coating could effectively improve the wear resistance of AZ91D magnesium alloy, and the CS-NCA coating showed best wear resistance. The reasons can be summarized as follows:

(a)    Compared with thermal spraying, cold spraying can obtain higher coating density.
(b)    The strong plastic deformation of the particles of the cold spray coating can increase the dislocation density [32–34], resulting in the dislocation of strengthening effect.
(c)    The deposition of particles during cold spraying causes a high residual compressive stress inside the coating [35,36].

Therefore, the CS-NCA coating had a higher hardness and showed superior wear resistance to the other two thermal sprayed coatings.

### 3.4. Electrochemical Results

Figure 9 shows the electrochemical test results of the substrate and three coatings in 3.5 wt.% NaCl solution. Figure 9a shows the trend of OCP over time. After being immersed for 1 h, the potential of the AZ91D magnesium alloy was stable at −1.56 V; the stable potentials of the APS-KF coating and the APS-NCA coating were similar (−0.89 and −1.00 V, respectively); while the stable potential of the CS-NCA coating was only −0.08 V. As the higher the OCP value, the harder the coating is corroded; hence, the CS-NCA coating was harder to be corroded in 3.5 wt.% NaCl solution.

The OCP curve of the substrate was relatively stable. This was because the AZ91D magnesium alloy formed an oxide film in the 3.5 wt.% NaCl solution corrosive medium; nevertheless, the oxide film was not dense enough to prevent the corrosion from continuing. The OCP curve of the APS-KF coating had a tendency to rise first and then fall within 300 s, which could be attributed to the presence of many pores and cracks in the coating, and a complete corrosion protection film could not be formed. The OCP curve of the APS-NCA coating showed a trend of continuous decline towards negative potential, which might be caused by the rupture of the oxide film on the coating surface and new surface exposure [37,38]. The OCP curve of the CS-NCA coating appeared to rise slowly for a long time; this was because the protective film on the surface of the coating was gradually complete, while the OCP value tended to be stable at the end of the immersion period, which was due to a complete and dense oxide film formed on the coating [39].

Figure 9b shows the changing trend of the polarization curve. It can be seen that the polarization behavior of the two thermal spraying coatings was very similar, and the polarization curve of the CS-NCA coating was quite different from that of the substrate. At the same time, the cathode extrapolation method was used to calculate the corrosion potential ($E_{corr}$), the corrosion current density ($I_{corr}$), and the Tafel slopes of the anode and cathode according to the potential kinetic curve, and they are listed in Table 4. It was calculated that the $i_{corr}$ value of AZ91D magnesium alloy was as high as $2637 \cdot \mu A \cdot cm^{-2}$, and the $i_{corr}$ values of the three coatings were $22.96\ \mu A \cdot cm^{-2}$, $26.98\ \mu A \cdot cm^{-2}$, and $0.04\ \mu A \cdot cm^{-2}$, respectively. Generally, the $i_{corr}$ value can be used to evaluate the active degradation rate of a material [40]. Therefore, the dissolution rate of the three coatings was much slower than that of the AZ91D magnesium alloy in 3.5 wt.% NaCl solution; moreover, the dissolution rate of the CS-NCA coating was the lowest.

In order to further understand the influence of the three coatings on the corrosion resistance of AZ91D magnesium alloy in 3.5% NaCl solution, electrochemical impedance spectroscopy (EIS) was measured at the stable OCP of the sample, and an equivalent circuit was made. Figure 9c–e represent impedance spectra, Nyquist diagrams, and Bode diagrams, respectively, and Figure 9f is a possible corrosion situation drawn based on the fitted equivalent circuit.

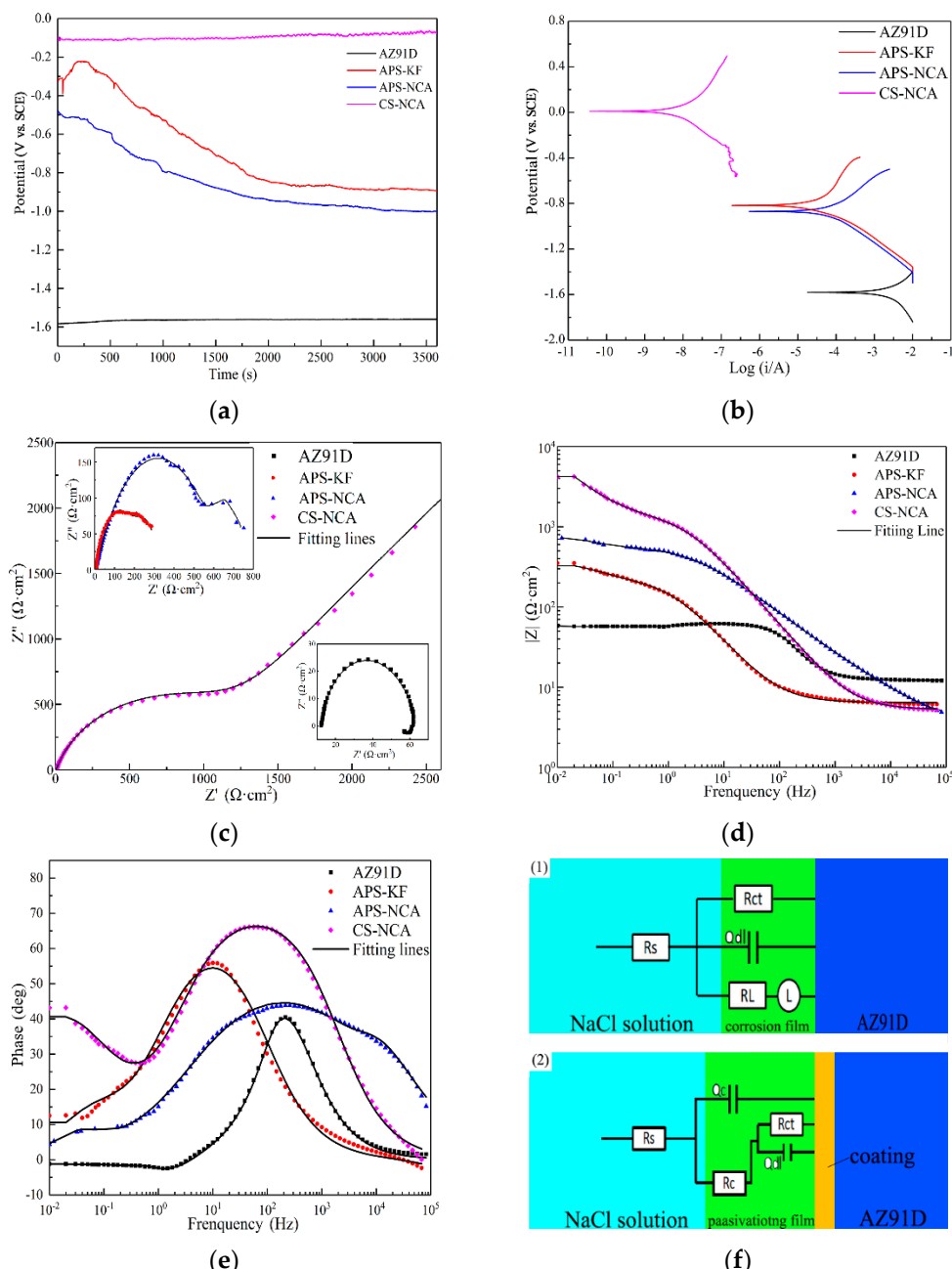

**Figure 9.** Electrochemical measurement results: (**a**) OCP vs. time curve; (**b**) PDP curve; (**c–e**) the Nyquist and Bode plots; (**f**) the EIS fitting equivalent electrical circuits.

**Table 4.** Polarization measurement results of the substrate and three coatings in 3.5 wt.% NaCl solution.

| Coatings | $E_{corr}$ (V vs. SCE) | $I_{corr}$ ($\mu A \cdot cm^{-2}$) | Anodic Tafel Slope (mV/dec) | Cathodic Tafel Slope (mV/dec) |
|---|---|---|---|---|
| AZ91D | −1.601 | 2637 | 201.49 | 256.93 |
| APS-KF | −0.823 | 25.96 | 219.58 | 55.682 |
| APS-NCA | −0.851 | 26.98 | 182.94 | 150.82 |
| CS-NCA | −0.02 | $4.404 \times 10^{-2}$ | 183.35 | 184.50 |

The impedance spectrum of AZ91D magnesium alloy was composed of a high-frequency capacitive loop and a low-frequency inductive loop. The electrical double



layer characteristics of the interface between the electrode and the solution generated a capacitive loop, and the dissolution of the passivation film in the solution showed an inductive loop [41]. Generally, the larger the diameter of the capacitor circuit, the lower the corrosion rate and the better the corrosion resistance. It can be seen in Figure 9c that the diameter of the capacitor circuit of the three coatings was significantly increased, and the capacitor circuit of the CS-NCA coating was larger than that of thermal sprayed coatings. In Figure 9e, it can be seen that the phase angle of AZ91D magnesium alloy was relatively gentle, and the phase angle was close to $-35°$ at mid and high frequencies. The phase angles of the APS-KF coating was $-40°$ in the low frequency, and the APS-NCA coating's phase angle was $-55°$ in the intermediate and high frequencies, respectively. In comparison, the phase angle of the CS-NCA coating was $-60°$ in the intermediate frequency. All of the three coatings showed broad crest. Therefore, the Nyquist diagrams of the three coatings showed capacitive loops of mid-high, mid-low, and mid-frequency, and there were diffuse components. Figure 9f is a circuit diagram of two actual possible circuits drawn according to the fitted equivalent circuit. Among them, $R_s$ was the solution resistance, $R_{ct}$ was the charge transfer resistance, $Q_{dl}$ was the internal electric double layer capacitance, $R_l$ was the passivation film resistance, and $L$ was the inductive impedance. Therefore, the equivalent circuit model of the AZ91D magnesium alloy can be expressed as $R_s(Q_{dl}R_{ct}(R_lL))$. The equivalent circuit model in Figure 9f (2) was often used to represent dense coating protection [42,43]. In the model, $Q_c$ represented the constant phase element of the coating passivation film, and $R_c$ represented the resistance of the passivation film, so the model could be expressed approximately as $R_s(Q_c(R_c(R_{ct}Q_{dl}W)))$.

Table 5 shows the fitting results of the two circuits to the EIS experimental data. It can be seen from the $R_S$ value that there was no significant difference in the solution resistance of the test sample, and $R_{ct}$ was an important parameter, indicating that the coating improved protection and was usually inversely proportional to the corrosion rate [41]. It can be seen that the $R_{ct}$ value of AZ91D was 91.43 $\Omega\cdot cm^2$ and that of the three coatings was 225.9 $\Omega\cdot cm^2$, 624.6 $\Omega\cdot cm^2$, and 900 $\Omega\cdot cm^2$, respectively. This indicated that spraying NiCrAl coating on the surface of AZ91D magnesium alloy could significantly increase the Rct value; moreover, the CS-NCA coating showed superior corrosion resistance to the other two thermal sprayed coatings.

**Table 5.** The calculated parameters of the equivalent circuit components of the substrate and the three coatings in 3.5 wt.% NaCl solution.

| Coatings | $R_s$ ($\Omega\cdot cm^2$) | $R_{ct}$ ($\Omega\cdot cm^2$) | $Q_{dl}$ ($Sn/cm^2$) | $R_c$ ($\Omega\cdot cm^2$) | $Q_s$ ($\Omega\cdot cm^2$) | $R_t$ ($Sn/cm^2$) | $L$ ($H\cdot cm^2$) |
|---|---|---|---|---|---|---|---|
| AZ91D | 12.19 | 91.43 | 0.87 | - | - | 119.5 | 30.39 |
| APS-KF | 12.8 | 225.9 | 0.79 | 87.19 | 0.97 | - | - |
| APS-NCA | 11.82 | 624.6 | 0.59 | 137.5 | 1 | - | - |
| CS-NCA | 10.55 | 900 | 0.79 | 11,440 | 0.75 | - | - |

In general, preparing NiCrAl coating by cold spraying process is a feasible method to improve the corrosion resistance of AZ91D magnesium alloy in 3.5 wt.% NaCl solution.

## 4. Conclusions

In this study, three kinds of NiCrAl coatings were prepared on the AZ91D magnesium alloy using plasma spraying process and cold spraying process, respectively. The structure, wear resistance, and corrosion resistance of the coatings were studied, and the conclusions are as follows:

(a)    The porosity of the two-thermal sprayed NiCrAl coatings is 3.21% and 2.66%, respectively, while that of the cold sprayed NiCrAl coating is only 0.68%; moreover, the hardness of the cold sprayed NiCrAl coating (650 $HV_{0.1}$) is obviously higher than those of the two-thermal sprayed NiCrAl coatings (300 $HV_{0.1}$, 400 $HV_{0.1}$).

(b) In the same sliding wear friction test environment, the three NiCrAl coatings have better wear resistance than AZ91D magnesium alloy; and among the three coatings, the cold sprayed NiCrAl coating has the lowest friction coefficient and the least wear amount. Therefore, the cold sprayed NiCrAl coating has superior wear resistance to the two-thermal sprayed NiCrAl coatings.

(c) Electrochemical experiments in 3.5 wt.% NaCl solution show that the corrosion current density of the two-thermal sprayed NiCrAl coatings is two orders of magnitude lower than that of the substrate, while the corrosion current density of the cold sprayed NiCrAl coating is two orders of magnitude lower than those of the two-thermal sprayed NiCrAl coating. Therefore, the cold sprayed NiCrAl coating has superior corrosion resistance to the two-thermal sprayed NiCrAl coatings.

**Author Contributions:** Conceptualization, G.L.; methodology, X.Z.; software, B.F.; validation, X.Z.; formal analysis, G.L.; investigation, X.Z.; resources, T.D.; data curation, Y.L.; writing—original draft preparation, X.Z.; writing—review and editing, T.D.; visualization, Y.L.; supervision, Q.L.; project administration, Q.L.; funding acquisition, G.L. All authors have read and agreed to the published version of the manuscript.

**Funding:** The work was financially supported by the National Natural Science Foundation of China under Grant No. 51675158.

**Institutional Review Board Statement:** Not applicable.

**Informed Consent Statement:** Not applicable.

**Data Availability Statement:** This research does not report any data.

**Conflicts of Interest:** The authors declare no conflict of interest.

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
