# Peer review of "Microstructure and Properties of Cold Sprayed NiCrAl Coating on AZ91D Magnesium Alloy"

_coatings, doi:10.3390/coatings11020193_

Round 1

Reviewer 1 Report

Dear Authors!

I appreciate your efforts in studying of "Microstructure and Properties of Cold Sprayed NiCrAl Coating on AZ91D Magnesium Alloy" and I recommend it for publication in MDPI "Coatings" after minor revision. You can find my comments in the word.doc attached. Besides I ask you to correct MANY typos in your manuscript such as missed spacings, wrong punctuation marks etc.

I have also very general comment - please, find a chance to discuss the issues of cost, technological simplicity and eco-friendliness of cold spraying in comparison with plasma spraying. 

Regards ...

Author Response

we responded to your comments in your attachment and revised in our manuscript.

Reviewer 2 Report

The reviewed work is an interesting aspect of the development of functional properties of magnesium alloys. Due to their low density, these alloys are a very attractive material, especially for the aviation and automotive industries. However, due to low mechanical properties and corrosion resistance, they are still a material of limited use. As shown in the work, the use of protective coatings can lead to an improvement in the performance of these alloys. The work was prepared carefully. Certain ambiguities can be found in several places. Comments are provided below. 1. There is no reference to Table 1 in the text. There is no information on what basis the chemical composition of the alloy and particles was determined. Was it based on the manufacturer's data? 2. What was the distance traveled against the specimen during the abrasion test? 3. Why the choice of NaCl with a concentration of 3.5% as the corrosive medium? 4. How was the porosity determined? Was the analysis carried out on the surface or across the section? What was the total area analyzed? 5. Please consider the graphic representation of hardness results. Wouldn't it be better to present the results with a broken line, fig. 4? 6. It is worth to include a photo of the imprint after the hardness measurement. 7. The thickness of the obtained layers was not analyzed. On what basis was it concluded that the cold coating line 231 was thicker?

Reviewer 3 Report

This paper describes a study of a cold-sprayed NiCrAl coating on a magnesium alloy, in which the abrasive wear performance and corrosion resistance has been evaluated and compared with other NiCrAl coatings deposited on the same magnesium alloy using plasma spraying. The results indicate that the cold-sprayed coating offers enhanced wear and corrosion resistance due to its higher hardness, lower porosity and lower friction behaviour.

The paper is generally well written and easy to read. However, there are a number of improvements that are needed before it can be accepted for publication. They are listed below.

  1. Introduction (line 28): I suggest the start of the sentence is changed to “Magnesium alloys have…”
  2. Introduction (line 30): please explain what “3C products” are.
  3. Table 2 seems to be missing some of the details in some of the columns. Also, why is the spraying temperature of the cold sprayed coating (CS-NCA) given as 850 °C when the process is “cold spraying”?
  4. Section 2.3 (line 99); the authors have quoted the hardness of the steel ball used in the abrasion test as 62 HRC. What is the equivalent Vickers hardness? I accept that conversion between different hardness scales is not straightforward; nevertheless, the authors could use a hardness conversion table to give the reader an approximate indication, particularly as the hardness values of the coatings are quoted in Vickers hardness.
  5. Table 3: what is the reasoning for choosing the abrasion test conditions?
  6. Section 3.1 (line 120): the figure numbering is incorrect (Figure 4 should be Figure 3).
  7. Section 3.1: what were the surface roughness values of the coatings? Were they tested in the as-deposited condition or did they have any post-deposition treatment such as grinding or lapping? Although the authors mention in Section 2.2 that the average surface roughness of the coating surface was measured, there doesn’t appear to be any mention of the as-deposited surface roughness and how the APS coating compares to the CS coating. In section 3.3 (line 162) the authors state that “…the APS-NCA coating was rough and uneven…” but provide no roughness values.
  8. Have the authors carried out any X-ray diffraction of the coatings? In some thermal sprayed coatings, there are different phases formed due to decomposition of starting materials. How do the plasma sprayed coatings compare with the cold sprayed coatings?
  9. Section 3.2: what were the standard deviations of the hardness values?
  10. Figure 5: it would be better if the authors used a different colour for the friction coefficient of the cold sprayed coating to distinguish it more easily from the APS coating. This also applies to Figure 9.
  11. Figure 6: please include a colour scale to indicate the depths of these wear scars more clearly.
  12. Section 3.3 (line 207): please change to “…the particles being well bonded…”
  13. Section 3.4 (line 269): how were the icorr values calculated?
  14. Section 3.4 (line 302): there is no Figure 10f.
  15. References: references [3] and [5] are the same.

Reviewer 4 Report

Dear Authors,

it is my pleasure to review your publication.

Here are some remarks:

Line 70: Missing blank in front of (c,d)

Line 73: Use Ø instead of Φ, which is unusual for the diameter

Table 1: Adjust to content -> this should improve the appearance

Chapter 2.3: How many measurements were carried out per coating? Only one?

Line 120 ff: The numbering Figure 4 is wrong. It is Figure 3.

Line 135: Figure 3: To show unmelted and deformed particles, pores and cracks please increase the magnification significantly

Line 151: Figure 4: Please insert error bars to the measuring points. Ist he hardness really uneven within the margin of error bars? The hardness in the case of APS-KF tends to decrease from the surface to the interface. Within the margin of the error bars?

Line 159: For my opinion a stable wear stage in the case of the APS-KF coating is reached at around 400 s not at 200 s.

Line 185: Repeating measurements can improve the findings and the uncertainty of the measurements

Line 192/ 193/ 197…: Please remain consistent with the designations such as figure 8 respectively Fig. 8

Line 248: … substrate is relatively…

Line 249: …an oxide layer…

Line 259: …oxide film was formed…

Line 261: missing blanks after numbering (a) and (b)

Figure 9b: Missing blank in the labeling of the ordinate. Maybe it would support the findings if the scaling of the ordinate is the same in Figure 9a.

Line 327: Missing blank.

Best regards

Reviewer 5 Report

The reviewed article deals with plasma and cold spraying Ni-Cr-Al coatings on magnesium alloy substrates. In general, the paper is well organized and its quality is good. Nevetheless, there are some remarks, which are listed below:

  1. Please use "wear trace" instead "wear scar".
  2. In the Introduction please add the one of the unique article, which deals with HVOF coating sprayed on magnesium alloy, e.g. Materials 2020, 13, 2775; doi:10.3390/ma13122775
  3. Are dimension of the powder particle from producer? Please add d50 (main diameter). Moreover, in case of KF-110 powder, dimension (20-60 µm) was which one (for elongate particles)?
  4. What was the surface roughness after sand-blasting?
  5. Table 1 - is it own chemical analysis?
  6. Line 79 - gases were instead gas was.
  7. Please check all superscripts and subscripts.
  8. Table 2 - what was the powder feed rate for Cold Spray process?
  9. Lines 93-95 - please explain clearly methodology of microhardness measurement.
  10. Table 3 - please add the sliding distance.
  11. Line 135 (caption) - Figure 3 instead Figure 4, please check and correct it also in the text.
  12. Porosity determination - please add the standard number (because it is standarized method). Moreover, these values are average ones, but where are standard deviation values? Please add it.
  13. Figure 4 (line 150) - please did not connect the points with lines, this is a scatter plot.
  14. Pleae explain, why CS-NCA coating exhibited lower CoF value, in compare to plasma sprayed ones and why,in your opinion, it was obvious?
  15. Lines 185-186 - are these values similar (8.85 and 4.698)? In my opinion wear lost for APS-NCA was closer to CS-NCA one, than to APS-KF one.
  16. For corrosion current density please check the unit.

Round 2

Reviewer 5 Report

All my remarks have been included.

Author Response

Thanks for your comments